# Neural Networks for Automatic Posture Recognition in Ambient-Assisted Living

**DOI:** 10.3390/s22072609

**Published:** 2022-03-29

**Authors:** Bruna Maria Vittoria Guerra, Micaela Schmid, Giorgio Beltrami, Stefano Ramat

**Affiliations:** Laboratory of Bioengineering, Department of Electrical, Computer and Biomedical Engineering, University of Pavia, 27100 Pavia, Italy; brunamariavitt.guerra01@universitadipavia.it (B.M.V.G.); micaela.schmid@unipv.it (M.S.); giorgio.beltrami@unipv.it (G.B.)

**Keywords:** human action recognition, ambient-assisted living, artificial intelligence, neural networks, machine learning, deep learning, feature selection, kinect, visual sensor-based

## Abstract

Human Action Recognition (HAR) is a rapidly evolving field impacting numerous domains, among which is Ambient Assisted Living (AAL). In such a context, the aim of HAR is meeting the needs of frail individuals, whether elderly and/or disabled and promoting autonomous, safe and secure living. To this goal, we propose a monitoring system detecting dangerous situations by classifying human postures through Artificial Intelligence (AI) solutions. The developed algorithm works on a set of features computed from the skeleton data provided by four Kinect One systems simultaneously recording the scene from different angles and identifying the posture of the subject in an ecological context within each recorded frame. Here, we compare the recognition abilities of Multi-Layer Perceptron (MLP) and Long-Short Term Memory (LSTM) Sequence networks. Starting from the set of previously selected features we performed a further feature selection based on an SVM algorithm for the optimization of the MLP network and used a genetic algorithm for selecting the features for the LSTM sequence model. We then optimized the architecture and hyperparameters of both models before comparing their performances. The best MLP model (3 hidden layers and a Softmax output layer) achieved 78.4%, while the best LSTM (2 bidirectional LSTM layers, 2 dropout and a fully connected layer) reached 85.7%. The analysis of the performances on individual classes highlights the better suitability of the LSTM approach.

## 1. Introduction

Human Action Recognition (HAR) aims to identify the actions performed by humans through the automatic analysis of data coming from different types of sensors. In the last years this method has become widely used in numerous relevant and heterogeneous application fields, from the most commercial to the most assistive ones, such as Ambient Assisted Living (AAL) [1,2,3,4,5,6,7]. In the latter, HAR provides an array of solutions for improving the quality of individuals’ life, allowing elderly people to live healthier and independently for longer, helping people with disabilities, and supporting caregivers and medical staff [3,6,7]. HAR is mainly based on the analysis of data acquired using several hardware devices (e.g., RGB cameras, RGB-D devices, or inertial sensors) and is carried out with a plethora of Artificial Intelligence (AI) algorithms [4,5]. The data acquisition for these purposes can be categorized into visual sensor-based and non-visual sensor-based (e.g., wearable inertial sensors), depending on which devices are used. The visual sensor-based approach using RGB-D systems, such as the Microsoft Kinect, allows collecting RGB, depth, and skeleton information data also providing a rich 3D structural information about the scene. Unfortunately, these data are frequently noisy and need pre-processing procedures for a robust estimation of the body position and the identification of human actions. Furthermore, the data quality is strongly dependent on the position of the subject with respect to the camera, on the movement complexity, and on the density of the furniture in the scene. More reliable data are obtained if the subject is facing the camera, or at least clearly visible in the camera optimal capture volume, with no body segments being occluded by the superimposition of other ones or any other object inside the room. A possible solution to handle these two latter constraints is a multiple camera setup to cover as many areas of the room as possible from different points of view.

As briefly mentioned above, the data for HAR are analyzed to classify and identify human actions with AI techniques, such as machine learning and deep learning models. The machine learning algorithms mostly employed are Support Vector Machine (SVM), Dynamic Time Warping (DTW), Hidden Markov Model (HMM), Random Forest (RF), and some kinds of Artificial Neural Networks (ANN) [5,8]. Malekmohamadi et al. compared the results of three different machine learning algorithms (Naïve Bayes (NB), Multi-Layer Perceptron (MLP) and RF) for identifying 13 possible human daily activities performed in front of the camera (i.e., standing, sitting lying down in sleep position, etc.), using Kinect skeletal joints’ coordinates. They obtained, respectively, an average precision value of 84.1% with NB, 98.7% with MLP, and 99.0% with RF [9]. Alternatively, Akyash et al. proposed a new kernel function based on DTW for SVM classification of eight different human postures (e.g., sit, walk, lay down, etc.) of two different data sets (TST fall detection dataset and UTD-MHAD dataset) [10,11]. The data of both datasets just mentioned were collected with the subject positioned in front of the camera. The proposed kernel was applied to each coordinate of every joint. With this method they obtained an overall accuracy classification of 98.8% with TST fall detection dataset and 98.75% with UTD-MHAD dataset [12]. Su et al. suggested a multi-level hierarchical recognition model, using a custom classification algorithm for processing Microsoft Kinect skeletal joints’ coordinates. At the first level, they used an SVM classifier, and at the second level, an HMM algorithm. With this solution, they aimed at identifying 20 human actions such as bend, hand catch, pick up, and throw, etc., using the MSRAction3D dataset [13], in which the actors were positioned in front of the camera during the acquisitions. They obtained an average recognition rate of 91.41% [14]. In the same vein, Ahad et al. trained a SVM classifier for human activities identification (e.g., walk, sit down, stand up, etc.) with kinematics features (3D linear joint positions and angles between bone segments) from a 3D skeletal joints datasets, in which subjects were positioned in front of the camera (UT-Kinect Action 3D, Kinect Activity Recognition Dataset, MSR 3D Action Pairs, Florence 3D, and Office Activity Dataset) [15,16,17,18,19]. The number of classes defined varied across 9 to 18, depending on the dataset used. The SVM classifier was trained with a linear kernel function obtaining, for each dataset, the following results in terms of accuracy and precision: 93.91%, 97.51%, 74.78%, 71.58%, and 94.92%, respectively [20].

Deep learning-based approaches are getting more attention in the HAR domain thanks to the progress they have made in terms of performance in the detection and recognition of human actions, especially in visual sensor-based studies regarding AAL environments. In particular, Convolutional Neural Networks (CNN) have achieved great success for image-based tasks, while Recurrent Neural Networks (RNN) outperformed other approaches on time series. For instance, Long Short-Term Memory (LSTM) networks are frequently used to solve sequence-based problems thanks to their strengths in modeling the dependencies and dynamics in sequential data [1,2,3,6,21,22]. Ahad et al. trained three different deep learning models using temporal statistical features computed through a sliding time window on 3D skeletal joints data from five public datasets and compared their performances with that of the SVM classifier. The first deep model was composed of two LSTM layers, the second one was arranged with one CNN layer followed by an LSTM network (CNNRNN), and the last model was organized with two CNN networks and an LSTM network for the last layer (ConvRNN). The best model for all the datasets used was the ConvRNN architecture, which obtained accuracies ranging 94.7% and 98.1% [20]. Zhu et al. proposed a new spatial model with end-to-end bidirectional LSTM-CNN (BLSTM-CNN). First, a hierarchical spatial–temporal dependent relational model was used to explore rich spatial–temporal information in the skeleton data. Then, a new framework was implemented to fuse CNN and LSTM. The LSTM was used to extract the temporal features and a standard CNN was used on the output of the LSTM to exploit spatial information. They used two well-known CNN architectures: VGG16 and AlexNET. The proposed models were trained and tested on the NTU RGB + D, SBU interaction, and UTD-MHAD datasets, and the number of classification labels ranged between 8 in the SBU Interaction dataset and 60 for the NTU RGB + D one [23,24]. In terms of overall accuracy, the BLSTM-CNN implemented with VGG16 provided the best results on the NTU-RGB + D dataset (87.1% and 93.3% in the cross-subject and cross-view benchmarks, respectively) and UTD-MHAD dataset (93.1%), while the AlexNET implementation was the best algorithm on the SBU Interaction dataset with 98.8% [25]. Alternatively, Devanne et al. compared two kinds of temporally hierarchical deep learning models to identify human activities of daily living through skeletal data captured with a Kinect V2 sensor. The first model was a conventional LSTM architecture with a single LSTM layer, a fully connected layer and a Softmax layer. The second one was similar but used an additional LSTM layer. They decomposed human activity sequences into a set of short temporal segments with the purpose of classifying 21 types of activity (10 human behaviors in a domestic environment and 11 in an office context). They obtained an overall accuracy of 58.9% regarding the domestic environment and 58.5% for the other one [26]. Zhu et al. proposed a deep LSTM network with three bidirectional LSTM layers and two feedforward layers. The last LSTM layer was a custom-designed LSTM layer, including dropout in order to prevent data overfitting. They trained and tested the classifier on three different online databases: SBU Kinect Interaction Dataset, HDM05 Dataset, and CMU Dataset [27,28]. Depending on the type of dataset used, they had a total of 8, 65, and 45 classes. They obtained an overall accuracy of 90.41%, 97.25%, and 81.04%, respectively, for each dataset [29]. Liu et al. proposed a tree-structure-based method to explore the kinematic relationship between the skeletal joints. They used these data as input to the first LSTM network layer, whose output was in turn fed to the second LSTM layer and finally a Softmax layer. In the two LSTM layers, a new gate was added to the LSTM block to handle the noise and occlusion in 3D skeleton data. They trained and tested this model with five different online databases (NTU RGB + D Dataset, SBU Inter-action Dataset, UT-Kinect Dataset, and MHAD) and obtained an overall accuracy of 77.7%, 93.3%,97.0%, 95.0%, and 100%, respectively, for each dataset [30]. On the other hand, Liu et al. proposed a new class of LSTM networks: Global Context-Aware Attention for skeleton-based action recognition, which was capable of selectively focusing on the informative Kinect joints in each frame by using a global context memory cell. The model is structured with a first LSTM layer, which encoded the skeleton sequence and generated an initial global context representation for the action sequence, and a second layer that performed attention over the inputs by using the global context memory cell. They trained the network on five different datasets, i.e., NTU RGB + D, SYSU-3D, UT-Kinect, SBU-Kinect, and MHAD, and achieved the following results in terms of accuracy: 76.1%, 78.6%, 99%, 94.9%, and 100%, respectively [31].

Working with high-dimensional data increases the difficulty of knowledge discovery and of pattern classification due to the presence of many redundant and irrelevant features. Dimensionality reduction of the problem, achieved by filtering or removing redundant and noisy information, allows to reduce or eliminate irrelevant patterns in the dataset, improving the quality of the data and, therefore, making the process of classification more efficient [32,33,34]. Feature selection is one of the techniques used to achieve dimensionality reduction by finding the smallest possible subset of features which efficiently defines the data for the given problem [35,36,37]. It can be accomplished using different methods, i.e., filter, wrapper, embedded, and the more recent hybrid approach [37,38,39]. The wrapper method selects the optimal features subset evaluating alternative sets by running the classification algorithm on the training data. It uses the classifier estimated accuracy as its metric [38]. The most used iterative algorithms are the Recursive Feature Elimination with SVM, the Sequential Feature Selection algorithm, and the Genetic Algorithm. Compared to the filter method, it achieves better performance and high accuracy [36,38]; nevertheless, it increases computing complexity due to the need to recall the learning algorithm for each feature set considered. Starting from the multimodal output of the Microsoft Kinect system, many sets of features of different nature (color-, depth- and skeleton-based) are used to train the classification models for HAR. To these aims, features usually range from RGB images, depth-based global features such as space-time volume, and silhouette information, to motion kinematic skeleton descriptors such as joint position and motion (velocity and acceleration), joint distances, joint angles, 3D relative geometric relationships between rigid body parts [40,41,42]. The possibility to fuse the different multimodal information obtained by RGB-D cameras has been recently explored giving good results [43,44].

In a previous study, we focused on skeleton-based features to classify the three most frequent postures taken by a person in a room during daily life behavior: standing, sitting, and lying down [45]. We have also considered a further posture, called “dangerous sitting,” representing a subject slumped in a chair with his/her head lying forward or backward as if unconscious. This allowed us to perform the first distinction between routine activities and alarm situations. Therefore, in order to develop a monitoring system able to deal with ecological data, we built a homemade database of skeleton data simultaneously acquired with four Kinect devices placed in different locations of the equipped room. In this way, data are as close as possible to the real daily scenarios in which the subject moves in the room, taking different orientations and different positions with respect to the camera. A subset of 10 features, computed from the Kinect skeletal joints coordinates and then chosen using the ReliefF algorithm, was used to train and to test a two hidden layers MLP neural network, obtaining, on the test set, an average posture classification accuracy of 83.9% [45]. This promising result was not, however, satisfying for our purpose since the classifier was the core of a more complex safety system aimed to generate an alarm when dangerous situations occur during everyday-life inside a room. Therefore, hoping to increase the MLP performance, in a later study, we proposed a pre-processing algorithm based on velocity threshold and anthropometric constraints [46]. In this case, the overall accuracy reached by the classifier was 92%, and it increased to 95% when the test data were also averaged in a timing window of 15 frames (corresponding to 0.5 s). This procedure, which performed so well, nevertheless had the weighty drawback of increasing the computational time considerably, making the process not useful for the online demand of the monitoring system. The time required for the pre-processing phase was about 1.031 s, and the frame-by-frame MLP classification was about 0.300 s when considering a sequence of 60 frames. At this point, two different developments could be attempted to improve the accuracy of the classification: 1. a computational optimization of the pre-processing algorithm previously described (not discussed in this context); 2. the tuning of the previously implemented MLP classification model and a search for a new model, more appropriate to manage the raw data noise and the constraints of an online safety system. This latter path is the aim of the present work in which, first, we defined a further class as the transition between two consecutive postures (for example, between sitting and lying down postures and vice-versa) to enable the system to handle the continuous stream of data from the Kinect device; second, we optimized the previous MLP neural network model [45] selecting a new subset of features using an SVM algorithm, a new set of network hyperparameters and a novel architecture; third, we trained and tested an LSTM sequence network model with a subset of features selected using a genetic algorithm. Both feature selection processes were carried out on the training data and started off from the previously selected set of 10 features [45]. We have chosen an LSTMs network expecting to take advantage of its ability to produce a frame by frame output yet based on a sequence of data instead of only the current input and thereby having the potential to catch a wide range of dependencies among them. Therefore, using this dynamic network we expect to also be able to classify the data referring to the transitions between two successive postures, e.g., when the subject passes from the standing to the sitting posture or from the sitting to lying posture and thereby configuring our system for usage in a more ecological daily life scenario in which the subject freely moves in the room. We have analyzed different LSTM sequence architectures to find the one that gives a higher level of performance. Each one has been configured to separately classify each data frame in the data sequence in order to have results comparable with those of the optimized MLP. The final step of this work was then to compare the performances of the two optimized algorithms.

## 2. Materials and Methods

### 2.1. Data Acquisition

As previously explained, the development of the intended monitoring system required a custom database of ecological skeleton data relative to subjects freely moving in the surveillance room for recognizing, in each data frame, the posture of the subject. No specific orientation of the subjects with respect to the Kinect systems was, therefore, required and held during data acquisition. We chose to separately classify individual frames in order to feed it to a multifactorial decision system taking into account the position of the subject in the room and relative to the furniture and data from other sensors and deciding whether to trigger an alarm. We, therefore, classified primitive postures instead of entire actions as mostly found in the HAR literature, and for this reason, we chose to collect an entirely new set of data. Such data were acquired and labeled during our previous work (see Material and Methods [45]). Briefly, we recorded 11 healthy subjects (7 females and 4 males; age ranging 25 and 60 years old; height ranging 1.55 and 1.90 m) while performing a total of 265 trials of about 13 min each. In each trial, subjects were asked to achieve an ordered sequence of postures (standing, sitting, lying, and slumping in a chair with the head leaned forward or backward), transitioning from 1 posture, lasting 10 s, to the following one without breaks. The subjects were simultaneously acquired by four Kinect One devices located in different sites of the experimental room (see Materials and Methods of [45]), and each acquisition was separately added to the database for training and testing the posture recognition model. All the participants gave written informed consent in accordance with the Declaration of Helsinki.

### 2.2. Data Processing

The acquired data were processed with the same techniques described in our previous work [45]. Specifically, the skeletal data of 17 joints (Figure 1) acquired by each Kinect V2 were roto-translated in a common, room-fixed, reference system and 16 joint angles plus 4 absolute angles were computed (Table 1). Of the three joint spatial coordinates only the vertical one (Z) was considered for further analysis and then normalized with respect to the height of the subject. All angles were normalized by dividing them by 180.

All missing data (temporal holes between data frames) were filled in with the value ‘999’. Each frame was labeled with the corresponding class among the 5 possible ones: standing posture (Class 1), sitting posture (Class 2), lying down (Class 3), “dangerous sitting” posture (Class 4), and transitions between 2 successive postures (Class 5). This latter class, not defined in the previous study, represents the transition between 2 consecutive postures (i.e., between a sitting posture to lying down posture and vice-versa or between a standing pose and sitting posture and vice-versa). It was added aiming to consider a more ecological experimental model, considering that monitoring subjects during their normal daily activities involve static and dynamic posture phases. We grouped together all the transitions in a single class in order to have sufficient frames and a balanced database for the training of the classifier. The frames containing the 999 values were labeled with the class selected for the last ‘not-999’ preceding frame (only for the LSTM database). All the described computations were performed using Matlab 2020a on an Intel i7 2.3 GHz quad-core architecture with 32 Gb of RAM.

### 2.3. Neural Networks

#### 2.3.1. Multi-Layer Perceptron

##### Feature Selection

Since, with respect to the previous study, we have considered another class (Class 5), a new feature selection to identify the best subset of features was performed. An SVM Classifier with a Gaussian kernel function was used to test all possible combinations of the 10 features selected in our previous work, for a total of 1023 possible combinations tested. For each combination, the loss of function was computed, and the combination of attributes with the lowest loss function was then chosen as the best set for the MLP classifier. All the feature selection process was carried out on the training set (divided into training—70% and validation set—30%).

##### Architecture

Starting from the MLP architecture implemented in the previous study [45] we investigated the effects, on the classification accuracy, of the number of neurons (for each layer we tested a number of neurons varying from 2 to 10) and the type of activation functions (Hyperbolic Tangent, Sigmoidal Tangent, ReLu, Pure Linear and, only for the last layer, Softmax) in order to achieve a final optimized MLP network.

The training database was then divided into training (70%) and validation set (30%), and the network was trained and validated for 10 times using a k-fold cross-validation (k = 10) for each set of considered hyperparameters. Once we selected the hyperparameters and after the validation process, the MLP was finally trained using the training database, first with a 10-fold cross-validation, and then using the whole training database. The learning process was per-formed over a maximum of 1000 epochs, i.e., 1000 iterations on the training database. The optimized architecture was finally tested on the testing database for 30 simulations for statistical analysis.

#### 2.3.2. LSTM Sequence Neural Network

Considering the addition of Class 5 with posture transition data, we speculated that recurrent neural networks (RNN) could perform better than static ones for their intrinsic nature of dynamic networks, i.e., systems whose output does not depend solely on the current input but also on the history of the input and of the network. We, therefore, chose to implement an LSTM network for posture classification and ran an optimization procedure to determine the best network hyperparameters for the task at hand. In order to be able to compare the results of the LSTM architecture with those of the MLP, we chose to configure the network to classify each input frame as the MLP does, i.e., an “LSTM sequence” network.

##### Feature Selection

A separate feature selection was performed for the LSTM sequence network as this recurrent network uses different principles for classifying the data, i.e., the network output depends on both the current input and on the state of the network, determined by the history of previous inputs. We then considered a standard LSTM sequence architecture similar to the one proposed by Devanne et al. [26] with an input layer followed by an LSTM sequence layer, a dropout layer, a second LSTM sequence layer, another dropout layer, a fully connected layer, and a Softmax layer as our reference architecture. Such network layout, with 100 and 75 neurons in the two LSTM layers and a 30% dropout layer, was used to compute the percentage of correctly classified postures (%CC), which represented the fitness function of the genetic algorithm that we used for the new feature selection. To this goal, we ran a custom-developed steady-state genetic algorithm working on a population of 50 binary chromosomes of 10 genes each (one for each feature). In the initial population, 49 chromosomes were randomly generated while one was forced to have all genes set to 1 in order to test considering all available features. The fitness function built a new LSTM sequence network for evaluating each chromosome, thereby exploring the different combinations of our 10 features, and each network was allowed for 20 epochs of training. The genetic algorithm was allowed to evolve for 10 generations. On the first run of the algorithm the genetic operators led to evaluate 390 different combinations of features (out of the possible 500, the remaining 110 being duplicates). The same 390 networks were then further trained 3 times for another 20 epochs in order to test their performance with different initial sets of synaptic weights and biases. The best result in terms of %CC or, in case of comparable %CC results, the set of the best results, identified the set, or the sets, of features, which we considered for the optimization of the network architecture (number of neurons in each LSTM layer and dropout percentage). This process of feature selection was computed only on the training set (Table 2).

##### Architecture

Since deep learning networks typically have many more parameters to learn than shallow networks and are trained over very large databases, the update of synaptic weights occurs after the network is presented with a subset of the available training data: a mini batch. The number of samples in mini batches influences learning in the network and represents, therefore, one of the hyperparameters to set. Moreover, when dealing with LSTM sequence networks, the input vectors were organized in sequences, typically of the same length. Choosing such length is another important hyperparameter to be considered. Therefore, based on evidence obtained in a previous exploratory analysis, in which we considered different length sequences of either 15, 30, or 60 frames (equal to 0.5, 1, or 2 s) and mini batches of 16, 27, 32, or 64 sequences, the sequence length was set to 60 frames, and the mini batch size was set to 32 sequences each.

The optimization process of the LSTM sequence network architecture was carried out selecting the number and position of dropout layers, the number of hidden LSTM layers and the number of neurons in each of them. We then explored the following architectures (in parentheses the acronyms used to indicate these architectures in the rest of our work): using only 1 LSTM layer (LSTM), 1 LSTM layer followed by 2 fully connected layers (LSTM2FC), 2 LSTM layers (2LSTM), 1 bidirectional LSTM layer (BLSTM), 2 bidirectional LSTM layers (2BLSTM) with 2 equal dropout layers, 2 bidirectional LSTM layers with 2 different dropout layers (2BLSTM2D), 2 bidirectional LSTM layers with only 1 dropout layer (2BLSTM1D). The number of LSTM hidden units varied between 75, 100, or 125 and those of the second layer, in the architectures that considered one, were 25 fewer. Each network configuration was trained for 50 epochs and its initialization and training were repeated 10 times for statistical analysis.

### 2.4. Database

The full database included a total of 734,339 frames, including those containing the 999 values. This latter was 35,020 and was discarded for the training and the testing of the MLP network. In this case, due to the static characteristic of the network, the temporal sequentially of the frames was not mandatory and to hold ‘999’ values added only noisy data. On the other hand, this removal has the limit to reduce the adherence of the model with the real scenario, corresponding to its final deployment setting, in which a proper reconstruction of the subject skeleton by the Kinect device is not always assured. The handling of the 999 values was thereby different for the LSTM sequence model where they were involved in the analysis. Only the sequences entirely composed by frames of 999 values were discarded. The database considered for the MLP network is thus composed of 699,319 frames, while the LSTM database includes 714,330 frames. The latter includes 15,011 (2.1%) missing data frames, i.e., ‘999’ frames, whose target class was set to the same class as that of the preceding frame. The class repartition of the 2 databases is shown in Table 2. A training set for the 2 networks was eventually built using the data from 10 of the 12 subjects, while the test set was built using the data of the remaining 2 subjects, as reported in Table 2.

A statistical analysis was performed on the results of the set of simulations that followed the 30 repetitions of the training of each considered network. The percentage of correct classifications obtained in each simulation was considered as a sample of distribution of results for the considered network, thus that the performances of the different networks could be statistically compared. The resulting sample distributions failed the test for normality of the data and were, therefore, compared using the Wilcoxon rank-sum nonparametric test. Statistical significance was considered for *p* values less or equal to 0.5.

## 3. Results

### 3.1. MLP Neural Network

The best set of attributes identified by the SVM feature selection algorithm was composed by five features: A_pitch, B_roll, Z_Head, Z_C7, and Z_Hc; which obtained a loss function equal to 0.293. This set of features was used for the training and testing of the MLP network.

As summarized in Figure 2, the MLP network architecture that had produced the best results, in terms of average accuracy, consisted of three hidden layers, each including 10 neurons with a Sigmoidal Tangent activation function. Moreover, the output layer, corresponding to the Softmax activation function, was composed of five neurons (equal to the number of classes in the database).

The overall total accuracy on the test set reached by 30 simulations of this MLP network architecture varied across 0.78–0.79 with a mean value of 0.784 ± 0.003 (Standard deviation (SD)). Figure 3 shows the distribution of the results of 30 simulations, separated for each class, for the Specificity, F-score, Sensitivity, and Precision statistical parameters. The Specificity results (Panel A, Figure 3), compared to that of the other three statistical metrics (Panel B, C, and D, respectively), were very close to each other and ranged 0.99–0.93. The F-score ranged from 0.83–0.91 in the first three classes showing a decrease in Class 4 (0.73), which became more significant in Class 5 (0.15). A similar behavior characterized the Precision results (panel D, Figure 3), where the values for Class 4 and Class 5 were equal to 0.71 and 0.09, respectively. These two values were very different with respect to the other three values varying from 0.90 to 0.95. In the Sensitivity results (panel C, Figure 4), the value of Class 5 was again the lowest (0.58) but much closer to the other four values, ranging between 0.74 and 0.88. To summarize, Class 3, corresponding to the lying posture, was the most precisely identified by the MLP classifier, followed by Class 2 (sitting posture) and Class 1 (standing posture). The network, on the other hand, classified Class 4 (“dangerous sitting” pose) and, especially, Class 5 (transition between one posture to another) with more difficulty for each of the four statistical parameters calculated (Figure 3).

Figure 4 shows the mean confusion matrix computed over the 30 network simulations performed. Observing the classification results, the major misclassifications were between Class 2 and Class 4 and vice versa (1.64% ± 0.12% and 4.95% ± 0.24%, respectively). Class 2 were mainly misclassified with Class 1 (1.54% ± 0.10%) and Class 4 were confused with Class 3 (0.60% ± 0.06%). The classifier worse performed in the identification of Class 5, which was confused with all the other four Classes (2.91% ± 0.06%, 4.40% ± 0.09%, 1.31% ± 0.05%, 2.07% ± 0.10%, respectively Class 1, Class 2, Class 3, and Class 4). The best-identified class was Class 2 (sitting posture), followed by Class 1 (standing posture).

The new model now classifies a sequence of 60 data frames in about 0.300 s.

### 3.2. LSTM Sequence Neural Network

As detailed in the Methods section, we used a two-step process for performing the selection of the best set of features for the LSTM network over the LSTM dataset, i.e., including ‘999’ frames. Each feature set proposed by the genetic algorithm was tested for computing its fitness by allowing the above-mentioned standard two LSTM layers network to train for 20 epochs. For statistical purposes, the same network was trained three more times for further 20 epochs; thus that we obtained four evaluations for each combination of features selected by a chromosome. The best four feature sets had similar fitness values and were then chosen based on the best performance over such four training trials. We finally performed one run al-lowing for 100 training epochs, which confirmed the four feature sets as the best-performing ones and, at the same time, allowed us to note a decrease in the percentage of correct classifications occurring after about 50 epochs with all feature sets, which was accompanied by an increase in the validation loss function. All four selected sets shared the first four features reflecting the roll and pitch angles of the trunk and of the head, two vertical coordinates among the three available, i.e., head, midpoint of the hips and midpoint of the shoulders, and two further angles. One set considered seven features, while the other three considered eight. These four sets were:Set 1: A_pitch, A_roll, B_pitch, B_roll, µ2, Z_Head, Z_C7;Set 2: A_pitch, A_roll, B_pitch, B_roll, ξ, µ2, Z_C7, Z_Hc;Set 3: A_pitch, A_roll, B_pitch, B_roll, µ2, δ2, Z_C7, Z_Hc;Set 4: A_pitch, A_roll, B_pitch, B_roll, ξ, δ2, Z_Head, Z_Hc.

With such optimized hyperparameters, the best performance achieved by the standard LSTM network with the four considered sets of features over the test set ranged from 81.9 to 83.5%. The latter was achieved using the Set 3 of features, which was the set chosen for the rest of our study.

Further search for the remaining network hyperparameters was then performed with the chosen feature set, exploring the layout of the network, the number of neurons in each LSTM layer, and the dropout per-centage, which were investigated considering 30 experiments, each one training the network for 50 epochs. All architectures performed best with a first hidden layer of 125 neurons and a second one of 100. The improvement in accuracy compared to the two layers having 100–75 and 125–75 neurons was in the order of 0.5% for both architectures, which was a consistent but not statistically significant finding.

The five dropout percentages from 25 to 45% considered for each tested architecture caused changes in the mean percentage of correct classifications in the order of 1%, which were not statistically significant. One example of the dropout results for the two LSTM layers and the one biLSTM layer networks is shown in Figure 5, considering the architectures having the highest number of neurons (*n* = 125).

A comparison of the accuracy results obtained with the different considered architectures is shown in Figure 6 for a chosen dropout level (45%). Bidirectional LSTM architectures achieve almost 2% higher accuracies than traditional LSTM layers even given a comparable number of neurons, while increasing the number of bidirectional LSTM layers does not significantly improve results.

The shuffling of the training sequences at every training epoch significantly improved the network performance compared to the no-shuffling procedure, as shown in Figure 7 for the two LSTM layers, the single and the two bidirectional LSTM layers architectures.

Overall, the best performance was, therefore, that of the 2 bidirectional LSTM layers network as detailed in Figure 8, with 125–100 neurons in the two hidden layers, two 45% dropout layers, one following each of the LSTM layers, trained with sequence shuffling at every epoch, which obtained a mean accuracy of 0.857 ± 0.008.

The network performance in terms of sensitivity, F-score, specificity, and recall for each of the classes is shown in Figure 9. The Specificity results (Panel A) show that the network rarely misassigns frames to the standing (Class 1) and sitting classes (Class 2) (mean 0.93 and 0.90, respectively), the lying posture (Class 3) has a mean specificity of 0.86, the dangerous sitting posture (Class 4) attains 0.85 and the transition class (Class 5) 0.72. The Sensibility of the network is its most appealing feature (Panel C), with values ranging from 0.91 to 0.98, with the dangerous sitting class specificity reaching 0.95. The F-score (Panel B) ranged from 0.90 to 0.95 in the first three classes, showing a decrease in Class 4 (0.88) and worsening for Class 5 (0.76). A similar behavior characterized the Precision results (panel D), where the values for dangerous sitting posture and the transition class were equal to 0.82 and 0.63, respectively, while the other three classes ranged from 0.84 to 0.92.

To summarize, Class 1, corresponding to the standing posture, was the most precisely identified by the LSTM classifier, followed by Class 2 (sitting posture) and Class 3 (lying posture). The network, on the other hand, classified Class 4 (“dangerous sitting” posture) and, especially, Class 5 (transition between a posture and another) with more difficulty for specificity, F-score, and precision (Figure 9).

The mean confusion matrix for such a network over the 30 training runs is shown in Figure 10.

Compared to the MLP, the misclassifications between Class 2 and Class 4 and vice versa decreased to 1.37% ± 0.35% and 2.46% ± 0.39%, respectively. Class 2 samples were also misclassified as Class 1 (1.16% ± 0.22%), while some Class 3 frames were classified as Class 2 (2.49% ± 0.18%). The frames that were erroneously classified as Class 5 were significantly fewer than with the MLP (0.59% ± 0.12%, 0.72% ± 0.11%, 0.63% ± 0.10%, 0.59% ± 0.13%, for Class 1, Class 2, Class 3, and Class 4, respectively). The best identified class was Class 1 (standing posture), followed by Class 2 (sitting posture).

The 2BLSTM2D model now classifies a sequence of 60 data frames in about 0.030 to 0.040 s.

## 4. Discussion

This study is part of a research project aimed at developing a monitoring system allowing frail individuals to live autonomously while being non-invasively monitored for the occurrence of dangerous situations, which may require external intervention. In a previous paper [45], we defined an MLP network aimed at classifying human postures of a subject performing daily living activities in a mock-up bedroom. Namely, we considered a set of features computed on skeleton data to recognize the ‘standing,’ ‘sitting,’ ‘lying,’ and ‘dangerous sitting’ postures. As previously described, our database was built using four different Kinect devices that simultaneously acquired the subject from four different points of view without a constraint on the position and the orientation of the subject with respect to the camera in order to train and test our model on more ecological data [47]. The present work develops on our previous paper, expanding the original dataset by including a fifth class, ‘transition’, collecting all transitions between two consecutive postures. We considered such a choice necessary in order to be able to provide the network with a continuous stream of data while reducing the risk of incurring in false positives during such transition movements, e.g., while sitting down. With such new database of classified postures, we performed a new feature selection on the 10 parameters that we chose to describe each captured frame in order to optimize the classification ability of an MLP network. The network hyperparameters were then, in turn, optimized, and we trained the MLP with our training set composed of the data relative to 8 of our 10 acquired subjects. The remaining two subjects’ data made up the test set, on which the MLP network achieved an average 78.4%CC.

The same five-classes dataset was considered to train an LSTM sequence network with the addition of 999 frames, i.e., frames in which the Kinect was not able to identify a proper skeleton. Such noisy frames were frequent within each acquired subject’s data, probably due to the varying orientations and positions of the subjects with respect to the Kinect systems. The presence of such noisy frames is a further element contributing to a dataset representing the intended deployment conditions of the overall monitoring system.

A new feature selection exploiting a genetic algorithm aimed at maximizing a fitness function consisting in the %CC computed over the test set using a reference LSTM sequence network was developed. Such network consisted in two LSTM layers, one 25% dropout layer, a fully connected layer, and a Softmax layer. The outcome of the feature selection led us to use 8 of the 10 available features, which were presented as sequences of 60 frames in mini batches of 32 sequences each. As above, the training set was then built using frames from eight subjects and the remaining to subjects’ data made up the test set. Several LSTM sequence architectures (one LSTM layer, two LSTM layers, one bidirectional LSTM layer, and two bidirectional LSTM layers), using different numbers of neurons in the hidden layer(s) and dropout arrangements, were tested in a hyperparameters optimization algorithm.

As expected, the addition of the ‘transition’ class worsened the classification ability of the MLP network. Indeed, frames that may be very similar to those that are required to be classified in one of the other classes are now being requested to be assigned to the ‘transition’ class. In other words, a transition between sitting and standing, for example, contains frames that are very similar to those belonging to the ‘sitting’ class and to those belonging to the ‘standing’ class. This can be appreciated by examining the last row of the confusion matrix in Figure 4, showing how a significant percentage of frames that were classified in Class 5 was supposed to be classified in the other classes. Clearly a static network such as the MLP has no means to correctly classify such frames.

The classification accuracy improved when using the LSTM architectures and more thus using bidirectional LSTM sequence architectures, in which half of the neurons are presented with the regular sequence of inputs while the other half is presented with the backward input sequence. Such network architecture, exploiting data shuffling and 45% dropout, reached the best results with a mean classification accuracy of over 85%. The effectiveness of this approach was especially evident both in terms of correct classifications for frames belonging to the ‘transition’ class, in which less than 3% of trials belonging to other classes are now classified (see Figure 7), and in terms of correct classifications of ‘999’ frames, for which the misclassified percentage was as low as about 9%.

Comparing the behavior of the two networks throughout the five classes shows a higher specificity for the MLP network on all the five classes, yet also a much lower sensitivity than the LSTM on each class. An important contribution to the overall higher %CC obtained by the LSTM is due to the improved ability to correctly classify frames pertaining to the ‘transition’ class, as mentioned, yet a significant improvement also regards the increased sensitivity, F-score, and precision in classifying the ‘dangerous sitting’ posture, which represents a critical condition for the real usage setting of the application, one requiring rising an alarm for the safety of the monitored user. From this standpoint the LSTM mean sensitivity of 0.95 in detecting true positives for the ‘dangerous sitting’ posture represents an important improvement from the 0.76 achieved by the MLP. Indeed, a higher chance of false positives (the specificity decreased from 0.93 for the MLP to 0.85 for the LSTM) represents an acceptable cost for being more certain that dangerous situations will not be missed.

In terms of computation times for data classification, the two suggested models (MLP, 0.3 s and BLSTM 0.03 s) were significantly faster than the previously suggested MLP with pre-processing procedure 1.331 s. However, the accuracy and sensitivity of the 2BLSTM2D model, which does not need a pre-processing phase, reached satisfactory results.

As a final consideration, the results that are reported in the literature and mentioned in the Introduction appear to achieve very high accuracy rates, typically around or over 90%. These are generally higher than those reported in our work and are obtained on a larger number of classes thus that a more in-depth understanding of these experiments in comparison to ours is called for.

The differences between these studies and the one presented in this work are broad and regard both the acquisition protocol, and hence the resulting database, and the goal of the classification approach. The studies reported in the literature aim at recognizing the daily action carried out in a sequence of data frames rather than facing the problem of recognizing one or more specific postures. Therefore, these works employ databases commonly available in the literature and adapt the set of actions to be classified according to the chosen database, without the need to create ad hoc ones. From this point of view, such studies are often focused on an artificial intelligence goal per se, although it may be tailored to specific aims such as monitoring the compliance with a rehabilitation or behavioral protocol, the identification of the daily living actions carried out to provide a measure of the subject’s daily activity, or for identifying changes with respect to his/her routine behavior. Our approach is instead quite specific, as it aims at recognizing individual postures during scenarios of everyday life, independently of the action that caused it (e.g., recognizing the lying down posture and not the falling action). In this setting, building our database using different camera points of view helps us to reproduce the data acquired on a subject freely moving in the room as during natural living conditions. However, this realistic approach increases the amount of noise in our data, threatening the accuracy of the classification results.

Altogether, these important differences lead us to consider our results as hardly comparable to those that can be found in the HAR literature. Nonetheless, we are aware that other deep learning architectures and approaches may lead to even better results than the ones presented in this work.

## 5. Conclusions

The best MLP model (3 hidden layers and a Softmax output layer) achieved an overall test accuracy of 78.4%, while the best LSTM (2 bidirectional LSTM layers, 2 dropout and a fully connected layer) reached 85.7%. The analysis of the results on individual classes highlights the better suitability of the LSTM approach, confirming the deep learning model capability to manage raw noisy data.

Summing up our results, the optimized 2BLSTM2D model represents a good compromise between the need for high overall classification accuracy and sensitivity for the dangerous sitting posture while compatible with a stream of skeletal data frames at 30 Hz.

Further developments for improving the performance of the system and its customization, for the specific purpose of allowing safer autonomous living conditions in frail individuals, may include the classification of each type of transition in a separate class and the use of LSTM ‘Last’ architectures. Indeed, for the ultimate patient-safety goal of the proposed system, a classification occurring every one or two seconds would not alter its effectiveness.

## Figures and Tables

**Figure 1 sensors-22-02609-f001:**
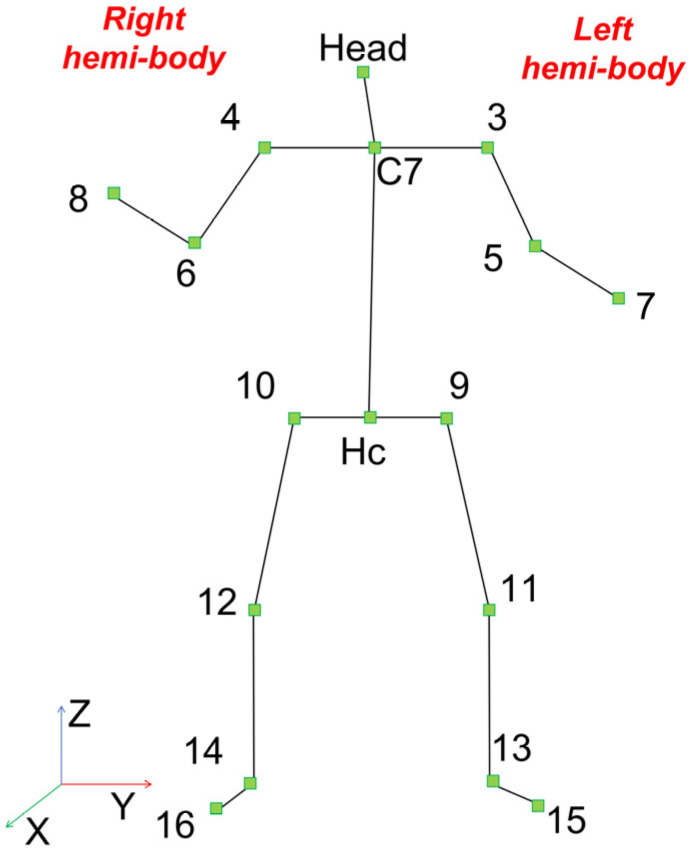
Skeleton with 17 joints.

**Figure 2 sensors-22-02609-f002:**
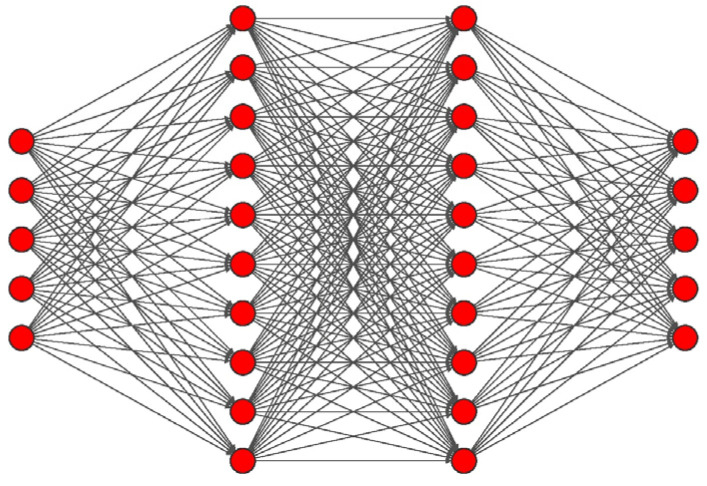
Optimized MLP architecture: 5 neurons in inputs and output, 10 neurons in the first, second and third hidden layers, each using the sigmoidal tangent activation function. The output layer was implemented with the Softmax activation function.

**Figure 3 sensors-22-02609-f003:**
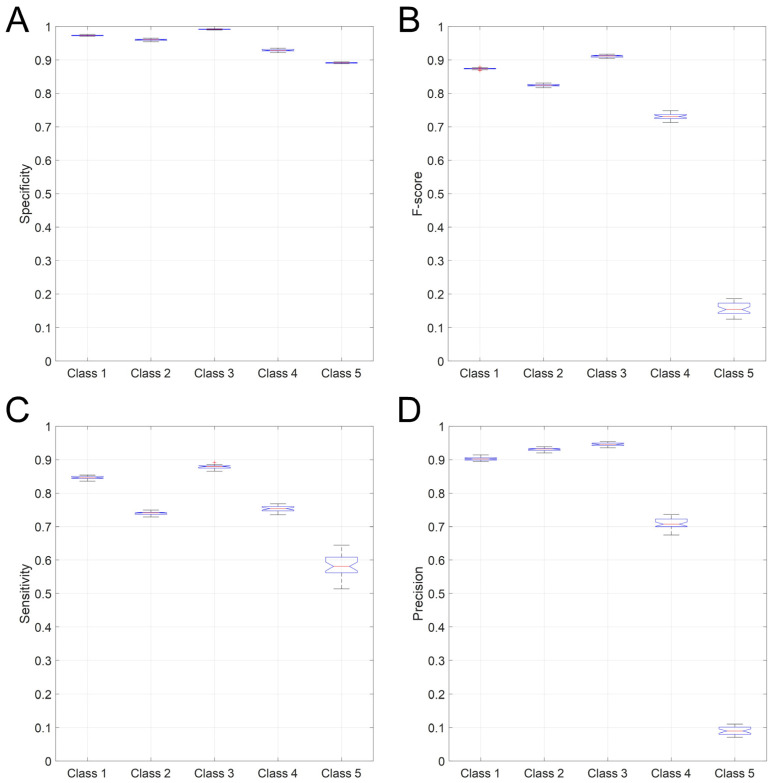
Specificity (panel (**A**)), F-score (panel (**B**)), Sensitivity (panel (**C**)), and Recall (panel (**D**)) are shown for the MLP optimized network 30 simulations. Each boxplot shows the median, the 25th and 75th percentiles, with the whiskers representing the maximum and minimum non-outlier values.

**Figure 4 sensors-22-02609-f004:**
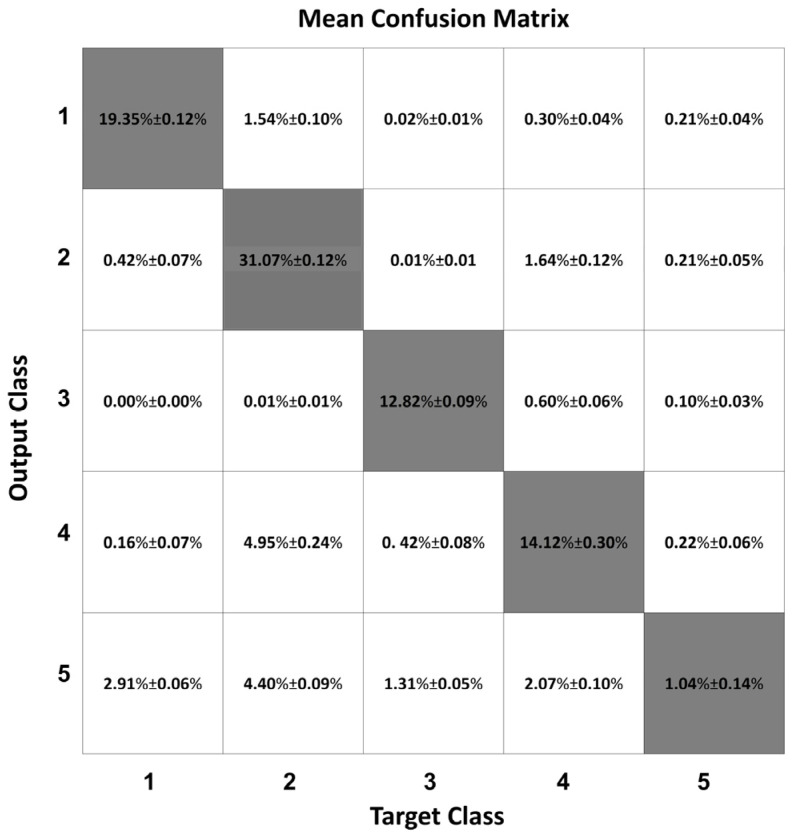
Mean confusion matrix obtained from the MLP network 30 simulations.

**Figure 5 sensors-22-02609-f005:**
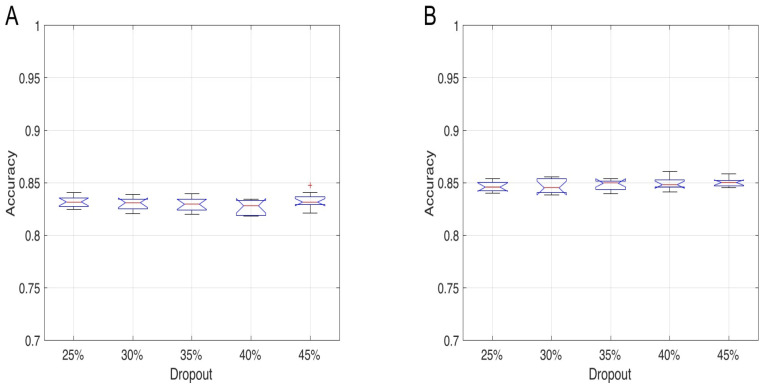
The overall test set accuracy distributions over 30 trained networks of two LSTM network architectures with different dropout levels. Panel (**A**): accuracy for a two LSTM layers network having 125 and 100 neurons in the first and second hidden layer, respectively. Panel (**B**): accuracy distributions for a one hidden bidirectional LSTM layer network with 125 neurons. Observations beyond the whisker length are marked as outliers and they are represented with “+” symbol.

**Figure 6 sensors-22-02609-f006:**
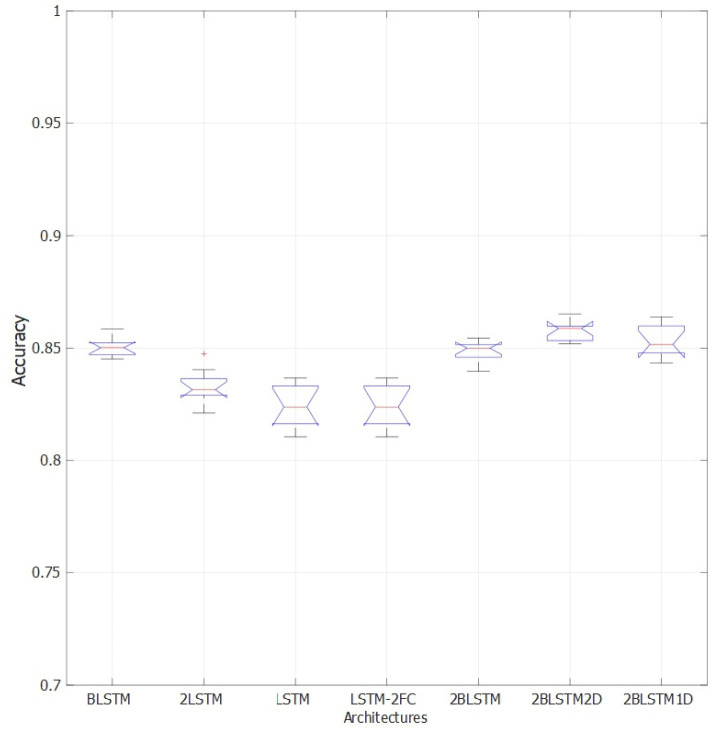
Accuracy distribution over 30 repetitions of 50-epochs training for different network architectures and dropout fixed at 45%. Tested architectures: one hidden bidirectional LSTM layer (BLSTM), two LSTM layers (2LSTM), one LSTM layer (LSTM), one LSTM layer followed by two fully connected layers (LSTM-2FC), two bidirectional LSTM layers (2BLSTM), two bidirectional LSTM layers with two dropout layers (2BLSTM2D), two bidirectional LSTM layers with one dropout layer (2BLSTM1D). Observations beyond the whisker length are marked as outliers and they are represented with “+” symbol.

**Figure 7 sensors-22-02609-f007:**
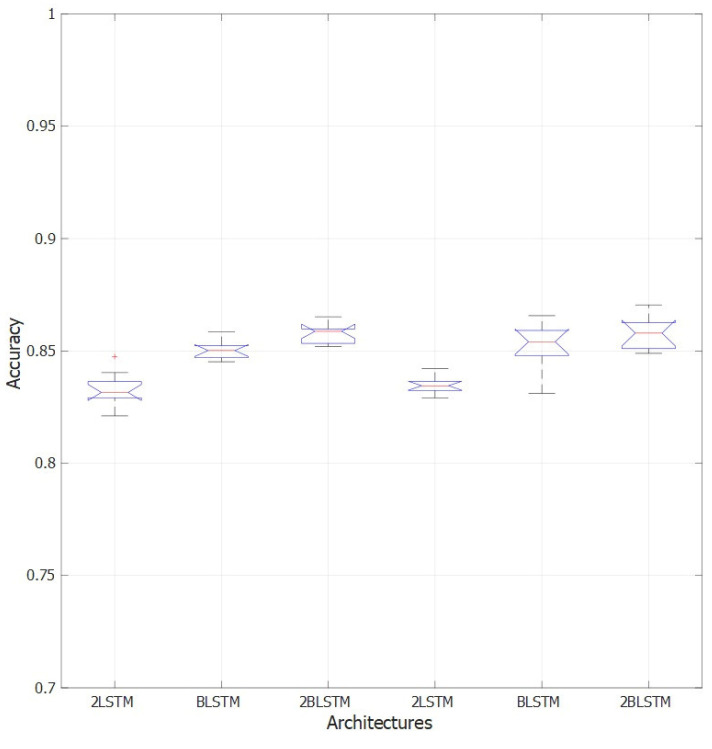
Effect of shuffling sequences at every epoch over the performance of the two LSTM layers network (2LSTM), the one bidirectional LSTM layer (BLSTM) and two bidirectional LSTM layers network (2BLSTM). The three leftmost boxplots represent the performance of the tested networks without data shuffling, while the three rightmost ones represent the performance of the same networks with data shuffling. Observations beyond the whisker length are marked as outliers and they are represented with “+” symbol.

**Figure 8 sensors-22-02609-f008:**
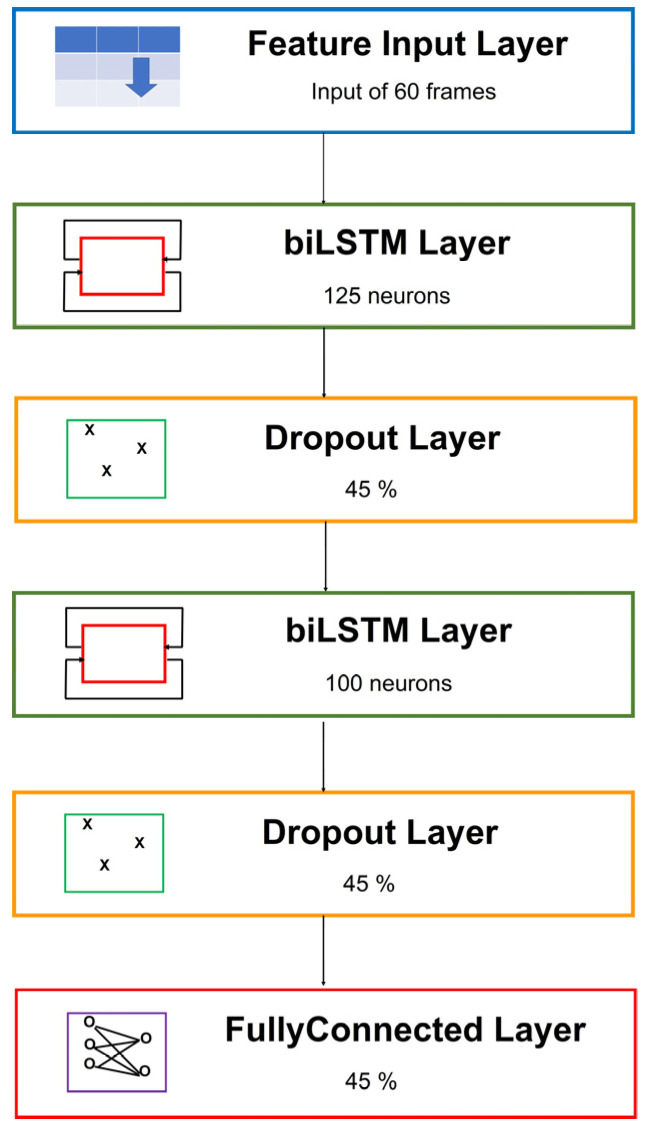
Definitive architecture of 2BLSTM2D model.

**Figure 9 sensors-22-02609-f009:**
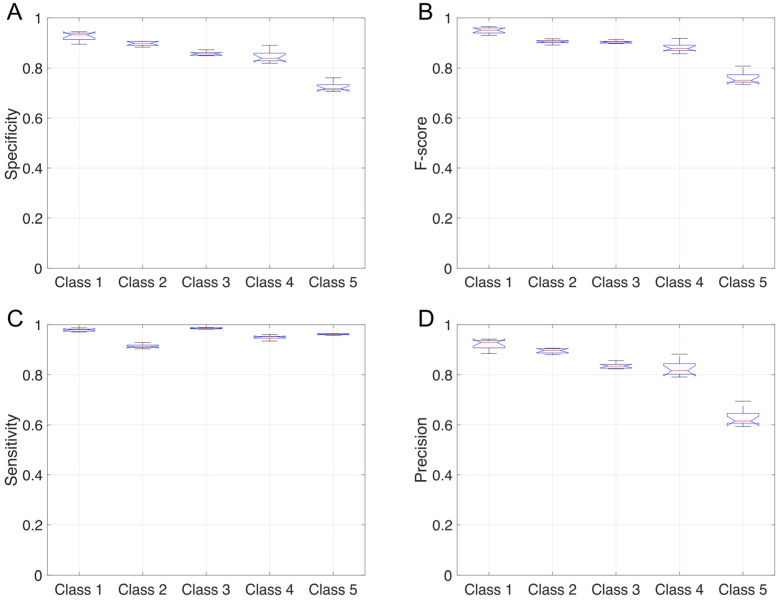
The four panels clockwise from the top present the Specificity (Panel (**A**)), F-score (Panel (**B**)), Sensitivity (Panel (**C**)), and Precision (Panel (**D**)) achieved for each class with the 2BLSTM2D network.

**Figure 10 sensors-22-02609-f010:**
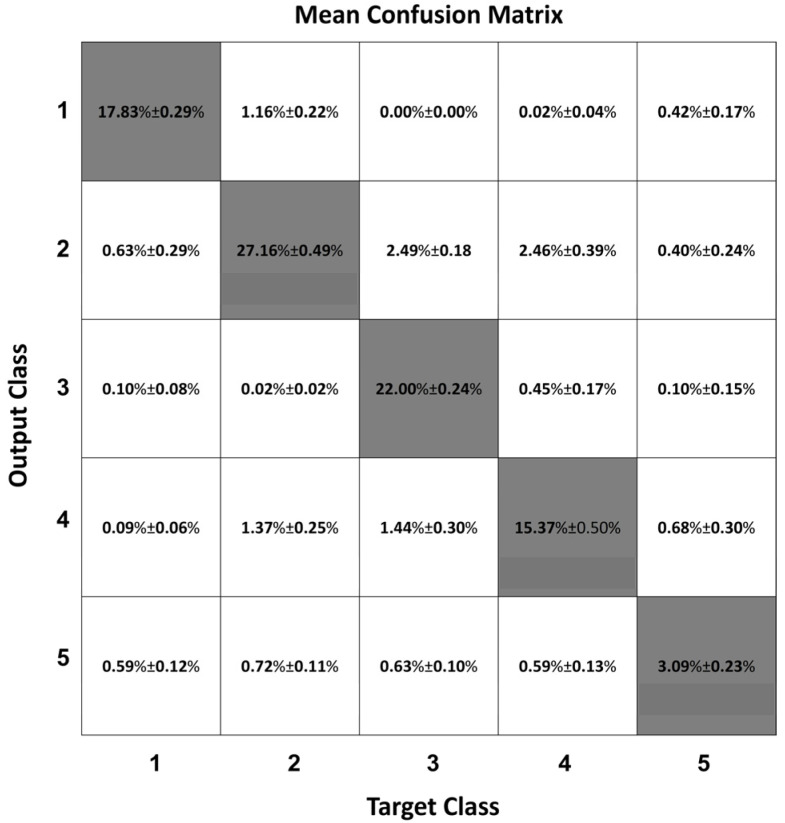
Mean confusion matrix obtained from the 30 simulations of the 2BLSTM2D network.

**Table 1 sensors-22-02609-t001:** Joint angles between two body segments are computed in the space, and absolute joint angles are calculated between a body segment and the horizontal plane passing through the two hips and the two shoulders.

Angles	Description
$\mu_{1},\mu_{2}$	Angle between head and shoulder segments. Left and right, respectively
ξ	Angle between head and trunk segments
$\tau_{1},\tau_{2}$	Angle between trunk and shoulder segments. Left and right, respectively
$\eta_{1},\eta_{2}$	Angle between shoulder and arm segments. Left and right, respectively
$\theta_{1},\theta_{2}$	Angle between arm and forearm segments. Left and right, respectively
$\delta_{1},\delta_{2}$	Angle between trunk and hip segments. Left and right, respectively
${ϒ}_{1},{ϒ}_{2}$	Angle between hip and thigh segments. Left and right, respectively
$\beta_{1},\beta_{2}$	Angle between thigh and leg segments. Left and right, respectively
$\alpha_{1},\alpha_{2}$	Angle between leg and foot segments. Left and right, respectively
$A_{roll}$	Head roll angle
$A_{pitch}$	Head pitch angle
$B_{roll}$	Trunk roll angle
$B_{pitch}$	Trunk pitch angle

**Table 2 sensors-22-02609-t002:** The numerosity of frames in the training and testing MLP and LSTM sequence databases and their repartition in five classes.

Classes	Database MLP		Database LSTM	
	Training	Testing	Training	Testing
Class 1	121,059	24,267	121,870	25,354
Class 2	189,668	43,990	193,446	45,258
Class 3	76,028	27,165	76,939	27,137
Class 4	113,178	23,779	116,903	24,838
Class 5	66,252	13,933	68,072	14,513
**Total**	**566,185**	**133,134**	**577,230**	**137,100**

## Data Availability

The data are publicly available here: https://github.com/bioingpv/Kinect-Database.git (accessed on 4 February 2022).

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
