# Peer review of "Neural Networks for Automatic Posture Recognition in Ambient-Assisted Living"

_sensors, 2022, doi:10.3390/s22072609_

Round 1

Reviewer 1 Report

Monitoring fragile people in their daily activities is the target of the paper. The authors use their own data set to discriminate between 5 classes of human postures, which include a posture considered as indication of a risk, and a class representing a transition between the others. Data are taken from 4 kinects in the room, which provide the main angles between the skeleton parts.

The aim of the authors is to improve their previously published model that uses the same data set, without considering the transition, using MLP. Moreover, the authors want their model to be compatible with the real time needs of a monitoring system.

They develop a MLP model, in a way similar to the previous experience, and then explore the use of LSTM network, make different optimizations. The final result shows that the net architecture containing bidirectional LSTMs has the best performance.

There are a few points to make clear.

- The state of the art (the second subsection of the Introduction) reports literature results in accuracy very high, and for much more classes than 5.

Have the authors considered in detail those solutions?

Are those results not reproducible?

As such, the subsection in practice asserts that good solutions are already available, and high performances have been already reached.

When reporting numbers it is important to indicate how they are obtained.

For instance, the accuracies are on test set?

Or on k-fold?

How many are the features considered?

What is the recognition time from the data acquisition to the classification output?

If the aim (as indicated by the authors) is to improve accuracy and computation time, the state of the art solutions should be reported on both.

Please rewrite this subsection to make the presentation more precise and to explain why the literature solutions have not been considered. Also explain why not to use a standard data set to make the results more easily comparable.

-In the Introduction please state in detail what are the targets of the new model, and why they are not yet fulfilled by the present methods.

- Section 2 explains the data set and the architectures developed for the classifier. Data have been divided into training and test set; the test set contains data of two persons not used for training. However, it is not specified whether or not the test data have been used in the process of feature selection. Another missing information is whether the very low number of features (5 to 7) selected does correspond to the ideal target of minimizing the feature number or is really obtained as the best.

- In Section 3 the authors report the results of the two models on the test set, after different optimizations. As expected, class 5 (the transition class) is quite often confused with all the other classes, as it contains positions very different.

The overall process of building the model takes a long time; however what is much more important is the execution time of the model. There is no mention about the computation times when applying the model. This is important. The authors only write at the end that monitoring every 1 to 2 seconds is acceptable. Please give the execution time as a whole or at least for feature calculation and model prediction.

Author Response

Dear Reviewers,

Thank you for giving us the opportunity to submit a revised draft of the manuscript “Neural networks for automatic posture recognition in Ambient-Assisted Living”. We appreciate you for your precious time in reviewing our paper and providing valuable comments. In the new version of the manuscript, we have incorporated the suggestions made by the reviewers. Please see the section below for a point-by-point response to the reviewers’ comments and concerns.

Thank you

Reviewers' Comments to the Authors:

Reviewer 1

  1. The state of the art (the second subsection of the Introduction) reports literature results in accuracy very high, and for much more classes than 5. Have the authors considered in detail those solutions? Are those results not reproducible?

As such, the subsection in practice asserts that good solutions are already available, and high performances have been already reached.

Thank you for pointing this out. The reviewer is correct but there are few reasons, now better explained in the manuscript (lines 169-174, 222-231 and 548-553) here below, which lead us to consider and to propose two new alternative solutions to the models described in the introduction. In the same vein the classification of a large number of actions or complex postures was out of our interest.

The models proposed in this work are part of a more complex monitoring system which needs the online postural information as a starting point for a multifactorial analysis. Very briefly, this latter integrates information about the posture, coming from the proposed model, with data coming from other environmental sensors and with the position of the subject in the room and with respect to the furniture. The smart combination of all these data is done to discern a dangerous situation from a normal one for the triggering of a warming alarm. Hence, the aim of the model is to recognize a restricted number of basic (primitive) postures (standing, sitting and lying down) which can belong to normal every day activities but that can also indicate the occurrence of dangerous situations: for example, the lying posture is part of a normal daily activity if taken on the bed, while on the contrary, it becomes a dangerous situation if it occurs on the ground. To these three general classes we added two further ones, specific to our project. One, “dangerous sitting”, for a first raw identification of an alarm situation, which grouped all the conditions of malaise or fainting, resulting in a seated person slumped or lying backward. The second one “transition” which identified the transition between two consecutive postures (e.g. between a sitting posture and lying down posture, and vice-versa, or between standing and sitting postures, and vice-versa) to deal with the flow of Kinect data and maintain the sequentiality of the postures. The solutions proposed in the literature and described in the introduction, mainly deal with the classification of daily living actions (e.g. drinking, reading a book on a sofa, falling down, ecc.), representing the single classifier output produced by a sequence of samples provided as input. The sequence of typically 50 to 150 frames is therefore classified as a whole, which is a different approach from the one explored in the present work, where each frame is being classified independently in terms of a posture instead of an action. Moreover, most of these models are trained and tested on data in which the subject is in front of the Kinect system or oriented at a specific camera angle. We think that these settings are suitable to obtain better quality data with limited noise, but they are not specifically ecological and therefore not so appropriate for applicative purposes such as ours.

  1. When reporting numbers it is important to indicate how they are obtained. For instance, the accuracies are on test set?
  2. How many features are considered?

We agree with the reviewer’s comment and we have now revised the manuscript to be more precise better specify these issues (lines 200-201). We started off with the 10 features that were chosen in our previous studies, then we selected 5 features for the MLP network (obtained with SVM algorithm) and 8 features for the BiLSTM model (chosen with the genetic algorithm). All the overall accuracies reported in the article are obtained on the test set.

  1. What is the recognition time from the data acquisition to the classification output?
  2. If the aim (as indicated by the authors) is to improve accuracy and computation time, the state of the art solutions should be reported on both.

Thank you for this suggestion. We have now added these information in the revised manuscript (lines 186 – 188, 421 and 523-524). Our goal is to reduce the computational time employed during the pre-processing phase suggested in our previous work (Guerra et al. 2020), which took about 1.031 seconds on an Intel I7 2.3GHz quad core computer to process a sequence of 60 frames. This was a further motivation to implement a LSTM deep learning model on top of the attempt to improve classification accuracy and being able to consider transitions between postures.

  1. Please rewrite this subsection to make the presentation more precise and to explain why the literature solutions have not been considered. Also explain why not to use a standard data set to make the results more easily comparable.

Following the reviewer’s suggestion, the subsection has been rewritten detailing the reasons leading us to exclude the models available in the literature as possible solutions. Moreover, we have explained why the definition of a new home-made database was mandatory (lines 169-174) to train and test our model. We are conscious that this choice makes it more difficult, if not impossible, to discuss our results in comparisons with those published in the literature, yet in real life the subject moves in the room in a random way without worrying about his/her body orientation with respect to the camera. Therefore, we have built a home-made database of data simultaneously acquired with four Kinect devices placed in different locations of the equipped room. In this way data are as close as possible to the real daily scenarios in which the subject moves in the room taking different orientations and different positions with respect to the camera. The data, coming from several points of view, with the subject not necessarily facing the camera and mimicking different sequences of postures, are noisy with a lot of missing data. This could be due to the fact that some joints overlap in some postures (for example, dangerous sitting one) or that the subject could exit from the camera sight causing temporary non-identifications of all skeletal joints. To maintain a good adherence with our final goal the model must be trained and tested taking into account all these real-world limitations.

a. In the Introduction please state in detail what are the targets of the new model, and why they are not yet fulfilled by the present methods.

Done (see the revised version of the Introduction)

b. Section 2 explains the data set and the architectures developed for the classifier. Data have been divided into training and test set; the test set contains data of two persons not used for training. However, it is not specified whether or not the test data have been used in the process of feature selection. Another missing information is whether the very low number of features (5 to 7) selected does correspond to the ideal target of minimizing the feature number or is really obtained as the best.

We thank the reviewer for the comment, now we have detailed the data used for both the feature selection procedures (lines 200-201). Moreover, the choice of starting the process of the feature selection from the reduced initial number of ten, excluding a large part of all the Kinect outputs, is due to previous reasoning described in our last paper used as a starting point for this one. We now have specified in the revised text that the feature selection was performed considering only the training data (lines 269-271).

c. In Section 3 the authors report the results of the two models on the test set, after different optimizations. As expected, class 5 (the transition class) is quite often confused with all the other classes, as it contains positions very different.

This is true for the MLP model, and it is one of the reasons for attempting a different solution based on a LSTM network, which indeed performs much better on this class (lines 590 - 595).

d. The overall process of building the model takes a long time; however what is much more important is the execution time of the model. There is no mention about the computation times when applying the model. This is important. The authors only write at the end that monitoring every 1 to 2 seconds is acceptable. Please give the execution time as a whole or at least for feature calculation and model prediction.

Thank you for pointing out the lacking information on computational times. The revised manuscript now details computational times for the preprocessing of a frame and its classification for both explored solutions (lines 186 – 188, 421 and 523-524).

Reviewer 2 Report

(1) The contribution of this paper is unclear in the current writing, especially when this paper is incremental work of an existing work of the authors' group.  For example, the literature review contains many references that are not closely relevant to this paper's specific contribution. It is not clear what challenges this paper has resolved that previous work could not. Moreover, the data capturing, feature selection, classifiers in the proposed method are mostly standard ML approaches and also very commonly used for HAR problems. As a result, the novelty of the proposed method is not convincing. 

(2)  The contribution of this paper has not been explicitly emphasized in the abstract, introduction, or conclusion sections, adding to the difficulty for the audience to find out which problem this paper has resolved that previous studies could not, and why the proposed methods could solve them. 

(3) The results reported classification error rates for different classes. However, it is not clear what those errors mean to the applications. Are "78.4%", "85.7%" good or bad results for the application?  Also, the experiments did not compare the proposed methods with sufficient state-of-the-art methods. 

Author Response

Dear Reviewers,

Thank you for giving us the opportunity to submit a revised draft of the manuscript “Neural networks for automatic posture recognition in Ambient-Assisted Living”. We appreciate you for your precious time in reviewing our paper and providing valuable comments. In the new version of the manuscript, we have incorporated the suggestions made by the reviewers. Please see the section below for a point-by-point response to the reviewers’ comments and concerns.

Thank you

Reviewers' Comments to the Authors:

Reviewer 2

  1. The contribution of this paper is unclear in the current writing, especially when this paper is incremental work of an existing work of the authors' group. For example, the literature review contains many references that are not closely relevant to this paper's specific contribution. It is not clear what challenges this paper has resolved that previous work could not. Moreover, the data capturing, feature selection, classifiers in the proposed method are mostly standard ML approaches and also very commonly used for HAR problems. As a result, the novelty of the proposed method is not convincing.

Thank you for pointing this out. In the revised manuscript we now highlight the contribution of our work. The goal of the proposed model is that of recognizing in each data frame the posture of a subject freely moving in a room, in order feed it to a multifactorial decision system taking into account the position of the subject in the room and relative to the furniture and data from other sensors, which may trigger an alarm for external intervention when a dangerous situation is identified. This is actually a quite different task from the one of identifying an action performed in front of the Kinect system, or at a specific orientation. We therefore aim at developing a proper body pose pattern recognition model, which is able to adapt to different daily living scenarios. Indeed, we decided to implement a deep learning model (LSTM), which is more suitable to handle noisy data and constraints related to a real time security system, while reducing the risk of incurring in false positives during such transition movements, e.g. while sitting down (Please see also Reviewer 1 section). The contribution of this work is both a database of real-world skeleton data acquired from several sub-optimal camera angles, and a deep learning network able to properly identify human postures in individual skeleton data frames, and to quite accurately recognize dangerous ones.

We have now reduced the number of citations from 55 to 46.

  1. The contribution of this paper has not been explicitly emphasized in the abstract, introduction, or conclusion sections, adding to the difficulty for the audience to find out which problem this paper has resolved that previous studies could not, and why the proposed methods could solve them.

We agree with the reviewer’s assessment. We have accordingly revised the abstract, the introduction and have added a conclusion section, in order to emphasize the paper contribution.

  1. The results reported classification error rates for different classes. However, it is not clear what those errors mean to the applications. Are "78.4%", "85.7%" good or bad results for the application? Also, the experiments did not compare the proposed methods with sufficient state-of-the-art methods.

In the light of our aim, the results are very good especially in terms of the LSTM mean sensitivity of 0.95 in detecting true positives for the ‘dangerous sitting’ posture, which represents an important improvement from the 0.76 achieved by the MLP. In fact, a high sensitivity on this class is critical for our goal, as it is the one representing the dangerous situation that the monitoring system needs to detect in order to trigger the alarm. The potential misclassification of other postures is less critical, and therefore tolerable.

Reviewer 3 Report

The paper presents an interesting methods, but in order to be a valuable research it needs seriously improvements:

  • it is not clearly what is the novelty of the paper; also what are the differences between previous paper?
  • what does it mean data captured by four Kinect de-vices? - data is acquired from 4 different viewpoints and after that they are merged? 
  • why was tested a MLP if currently all solutions are based on convolutional neural networks? Some benefits from the currently state of the art must be provided.
  • comparison with other existing methods must be added
  • why other existing datasets were not used for evaluation?
  • Kinect de-vices -> Kinect devices  

Author Response

Dear Reviewers,

Thank you for giving us the opportunity to submit a revised draft of the manuscript “Neural networks for automatic posture recognition in Ambient-Assisted Living”. We appreciate you for your precious time in reviewing our paper and providing valuable comments. In the new version of the manuscript, we have incorporated the suggestions made by the reviewers. Please see the section below for a point-by-point response to the reviewers’ comments and concerns.

Thank you

Reviewers' Comments to the Authors:

Reviewer 3

  1. it is not clearly what is the novelty of the paper; also what are the differences between previous paper?

We agree with the reviewer’s comment, for that reason in the revised manuscript we have now better emphasized the novelty of the paper and the differences with the previous one.

  1. what does it mean data captured by four Kinect de-vices? - data is acquired from 4 different viewpoints and after that they are merged?

Yes, data is acquired from four different viewpoints and added to a single database. Following your suggestions, in the new version of the manuscript we added more details about the acquisitions’ protocol (lines 237-240).

  1. why was tested a MLP if currently all solutions are based on convolutional neural networks? Some benefits from the currently state of the art must be provided.

In our previous work we implemented a MLP model to classify each data sample separately, while in this new study the goal is the tuning of the MLP hyperparameters and its comparison to a new solution. Since we are using temporal data, more precisely skeletal joints coordinates and angles, we decided to employ as a first deep learning model an architecture based on a recurrent neural network, such as the LSTM ‘Sequence’. Indeed, our type of classification is made frame by frame (in order to compare the LSTM results with the outputs obtained with MLP neural network). The reason why we decided to employ a LSTM model, instead of a CNN, is that the first one has the capability to keep in memory the full sequence of data provided, exploiting the long-term information to output a more accurate classification.

  1. comparison with other existing methods must be added.

The comparisons of our results with those of the literature is quite difficult because the dataset used to train and test both the models has been homemade and is relatively different from those of the public datasets. As specified to the Rev1 (question 6), the choice to use our dataset was required by the final purpose of the classification. The optimized model, proposed in this study, is used as an analysis module of a new safety domotic system, and as such must be defined on real data in terms of noise, missing data, shooting perspective of the Kinect camera, ecc. We have now better motivated why our results are little discussed in comparison with those in the (lines 222-230 and 548-553).

why other existing datasets were not used for evaluation?

For this answer, please see the answer to question 6 from Reviewer 1.

Reviewer 4 Report

The authors proposed a monitoring system detecting dangerous situations by classifying human postures through Artificial Intelligence (AI) solutions. The paper is interesting, but it needs a large formatting as it has large paragraphs. Also, the subtitles in the introduction are not needed, and more recent references are needed. The methods and results are interesting, but the statistical analysis of the results is needed. In the discussion, please compare with the previous literature, and please add the conclusions section with a summary of the conclusion.

Author Response

Dear Reviewers,

Thank you for giving us the opportunity to submit a revised draft of the manuscript “Neural networks for automatic posture recognition in Ambient-Assisted Living”. We appreciate you for your precious time in reviewing our paper and providing valuable comments. In the new version of the manuscript, we have incorporated the suggestions made by the reviewers. Please see the section below for a point-by-point response to the reviewers’ comments and concerns.

Thank you

Reviewers' Comments to the Authors:

Reviewer 4

  1. The authors proposed a monitoring system detecting dangerous situations by classifying human postures through Artificial Intelligence (AI) solutions. The paper is interesting, but it needs a large formatting as it has large paragraphs. Also, the subtitles in the introduction are not needed, and more recent references are needed. The methods and results are interesting, but the statistical analysis of the results is needed. In the discussion, please compare with the previous literature, and please add the conclusions section with a summary of the conclusion.

We thank the reviewer for this comment and have now revised the manuscript, accordingly, removing the subtitles from the introduction and adding a conclusion section.

For what concerns the recent references we respectfully disagree. In fact, about half of our references (about 20) are more recent than 2018.

The statistical analysis of the results was carried out considering the 30 training repetitions and corresponding simulations of the explored models as a sample of the distribution of the results and the different models were compared based on the nonparametric Wilcoxon rank sum test. This explanation has now been added to the Materials and Methods section of the revised manuscript (lines 367-373).

For the data comparisons with those of the previous literature, please refer to the answer to question 6 from Reviewer 1. We have now clarified this point (lines 229-232 and 548-553).

Round 2

Reviewer 1 Report

The authors have considered the indications from the first review and added the missing information.

In the answers to the reviewer they have explained the motivations and the target of the present paper, with considerations not fully reported in the amended paper. However, the present version clearly states the design and the result of the proposed method, which is part of an ongoing larger project.

Author Response

Dear Reviewer,

thank you for you comment.

Best regards,

Stefano Ramat

Reviewer 2 Report

No further comments. 

Author Response

(The authors gave the same response as above.)

Reviewer 3 Report

Some of my comments were addressed. But I consider that the proposed method must be compared with other existing ones (advantages / disadvantages, scenarios that can be used, etc)

Author Response

Dear Reviewer,

Thank you for you comment.  Please check the revision of our manuscript in which we hope to have answered your concerns regarding the comparison of our work to those in the existing literature at the end of the Discussion section. The revised text is reported in red.

Best regards,

Stefano Ramat

Reviewer 4 Report

The major comments are fixed, and it can be accepted as it is.

Author Response

(The authors gave the same response as above.)
